# High-Acceleration Vibration Calibration System Based on Phase-Locked Resonance Control

**DOI:** 10.3390/s22197208

**Published:** 2022-09-23

**Authors:** Ran Cheng, Zhihua Liu, Guodong Zhai, Qi Lv, Ming Yang, Chenguang Cai

**Affiliations:** 1School of Mechanical, Electronic and Information Engineering, China University of Mining and Technology, Beijing 100083, China; 2Division of Mechanics and Acoustics, National Institute of Metrology of China, Beijing 100029, China; 3Electrical Engineering College, Guizhou University, Guiyang 550025, China

**Keywords:** vibration control, phase resonance, vibration calibration, high acceleration, resonant beam

## Abstract

In order to ensure the measurement accuracy of high-acceleration vibration sensors used in engineering applications, it is necessary to calibrate their key performance parameters at high acceleration. The high-acceleration vibration calibration system produces high-acceleration vibration by utilizing the resonance amplification principle; however, the resonance frequency of the resonant beam changes with increasing amplitude, affected by the influences of nonlinear and other factors. In this study, a phase-locked resonance tracking control method based on the phase resonance principle is proposed to accurately and quickly track the resonance frequency of the resonant beam, which can improve the accuracy and stability of resonance control. The resonant beam is able to produce stable vibration with an amplitude exceeding 7500 m/s^2^ by phase-locking and tracking the resonant frequency. A calibration system built with this method can provide stable vibration with an amplitude of 500–10,000 m/s^2^ in the range of 80–4000 Hz. Comparison experiments with the commonly used amplitude iteration amplification method demonstrate that the proposed method can give an acceleration stability control index of less than 0.5% and a resonance tracking time of less than 0.1 s.

## 1. Introduction

As an indispensable and vital component for monitoring vibration and shock, high-acceleration sensors are widely used in different vehicles and industries [1,2]. For example, high-acceleration sensors are used to measure the vibration of various aircraft and spacecraft during their operation and to further prevent certain thin plate structures from being damaged due to high-acceleration vibration [3,4,5,6]. However, these sensors also need to be calibrated. Calibration of the sensors using a vibration table is an effective strategy to ensure their performance [7,8]. Therefore, it is imperative to develop an applicable vibration calibration system, applying a stable vibration control method to determine the performance of high-acceleration sensors accurately [9].

Currently, the maximum acceleration obtained by direct-push vibration-generating devices generally does not exceed 1000 m/s^2^ [10]. To obtain higher-acceleration vibration, mechanical structures are typically used to amplify the magnitude of the vibration. For example, shock vibration table testing technology was researched by Tohoku University in Japan [11], and the “Strong Vibration Generator” (SSG) was developed by the Central Research Institute of the Central Electric Power Industry in Japan [12]. 

However, there are some problems with the existing high-acceleration vibration calibration methods. The mechanism used for gain amplification is often a simple machine [13], and the resonance frequency of the resonant beam changes with increasing amplitude, affected by the influences of nonlinear and other factors [14]. The excitation vibration at a fixed frequency will no longer match the natural frequency of the mechanism after the amplitude changes. This results in a change in the magnitude of the resonance gain.

As shown in Figure 1, the resonant peak shifts in a fixed direction as the input amplitude changes. This is because the resonance frequency of the resonance mechanism changes with the change of the working acceleration. Resonance frequency change can be suppressed with the application of new materials, such as metamaterials [15,16,17]. However, under current conditions, the experimental system needs to adjust the excitation signal to ensure that the vibration is always in a high-acceleration state. Existing methods adjust the excitation amplitude at a fixed frequency. However, the amplitude of the resonant excitation at a fixed frequency above 7000 m/s^2^ becomes unstable and difficult to control [18]. This results in insufficient calibration capability of current vibration sensors under high acceleration.

With the vibration performance of commercial sensors gradually reaching or exceeding 10,000 m/s^2^ [19], there is an urgent need to improve calibration capacity to meet commercial and academic needs. In order to accurately and quickly track the resonant frequency of the resonant beam, a closed-loop calibration system is required for high-accuracy metrology [20].

Many research results on resonance control have used the phase resonance method due to its universality, high accuracy, and extensive redundancy [21]. In order to realize calibration of the sensor under high acceleration, the limitation of the vibration generation control device of the existing cantilever beam resonance amplification mechanism is considered [22]. In this article, referring to the control performance of different beams [23], the resonant mechanism is improved by adopting a fixed–fixed beam [24]. By analyzing active control of the resonant beam [25,26] based on the fundamental principle of phase resonance, the system resonant frequency as a function of acceleration is identified and tracked [27]. Phase-locked resonance tracking is used to control the vibration device to improve the controllability and stability of the resonance mechanism under maximum acceleration [28]. The experimental results show that the acceleration of the vibration calibration system can remain stable at more than 10,000 m/s^2^. The system can reduce the influence of frequency offset and enhance controllability when the system is excited at the highest acceleration.

The organization of this article is as follows: Section 2 introduces a high-acceleration vibration calibration system based on resonant amplification and expounds a phase-locked and stable-amplitude method for a phase-resonant system. The experimental results and a discussion are presented in Section 3, and Section 4 concludes the paper.

## 2. Materials and Methods

### 2.1. Resonance Calibration Device

Figure 2 shows the composition of the resonant amplifier IF calibration device (Dongling Co., Ltd., Suzhou, China) [29]. The standard sensor and the calibrated sensor are connected “back-to-back” as a comparison method for calibration [30]. The signal source sends out a sweep signal to determine the resonant frequency of the resonant beam test length [31].

The resonant mechanism can be viewed as a simple mechanical mass–spring–damper system [21].

Figure 3 shows the displacement equation of motion of the resonance mechanism is:(1)$$m\ddot{x}+c\dot{x}+kx=F$$
where *m* is the mass, *k* is the spring, *c* is the damping, *x* is the displacement of the resonating beam, and *F* is the exciting force of the vibration exciter.

The frequency characteristic function of the resonance mechanism is:(2)$$\frac{x\left(j\omega \right)}{F\left(j\omega \right)}=\frac{1}{m{\left(j\omega \right)}^{2}+cj\omega +k}=\frac{1/m}{{\left(j\omega \right)}^{2}+2\xi \omega_{n}j\omega +\omega_{n}^{2}}$$

The vibration amplitude is measured in acceleration; therefore, the characteristic function is converted to acceleration. Therefore, the expected transfer function *G(s)* of the resonance mechanism from the excitation force to the output acceleration should be:(3)$$G\left(s\right)=\frac{a\left(s\right)}{F\left(s\right)}=\frac{x\left(s\right)\cdot s^{2}}{F\left(s\right)}=\frac{s^{2}/m}{s^{2}+2\xi \omega_{n}s+\omega_{n}^{2}}$$

In the formula, ξ is the equivalent damping ratio coefficient of the resonance mechanism, $\xi =c/2\sqrt{km}$, and $\omega_{n}$ is the natural frequency of the resonance mechanism, $\omega_{n}=\sqrt{k/m}$.

Figure 4 shows the amplitude–frequency characteristic curve and its transfer function fitting curve of a certain length under the fixed input excitation signal of the vibration exciter obtained based on the open-loop manual control test method. The acceleration transfer function fitting the experimental data is:(4)$$Ga\left(s\right)=G\left(s\right)\cdot Gu\left(s\right)=\frac{s^{2}/m}{s^{2}+2\xi \omega_{n}s+\omega_{n}^{2}}\cdot ku=\frac{3.223\times 10^{4}\times s^{2}}{s^{2}+4.184s+5.853\times 10^{5}}$$
where *G_a_(s)* is the transfer function from the input voltage of the power amplifier to the vibration acceleration of the resonant output, we assume that the transfer function *G_u_(s)* of the power amplifier is the gain *k_u_*.

The form of Equation (4) conforms to that of Equation (3), which verifies that the gain mode of the mechanical structure conforms to the expected model. At the resonance mechanism’s natural frequency, the resonance mechanism’s magnification is $A(\omega_{n})=1/\left(2\xi \right)$, and the phase difference above and below the resonance mechanism is $\varphi (\omega_{n\varphi })=90{}^{\circ}$. Therefore, the smaller the equivalent damping ratio coefficient, the greater the amplitude gain of the resonance mechanism.

The 3D model of the resonant beam mechanism used in the study is shown in Figure 5. The material data used in the beam part are shown in Table 1.

At the same time, according to the above analysis, controlling the phase difference between the resonant beam and the vibrating table at around 90° can maintain the resonant beam mechanism in the optimal amplitude gain state. This is the basic principle of the phase resonance method.

### 2.2. Phase-Locked Tracking Control

The vibration control system studied in this article is realized based on the principle of phase resonance, and its tracking process is shown in Figure 6.

The basic principle of the phase resonance method is to assume that the modal parameters of the time-varying structure do not change within a short time interval. The measured phase difference between the excitation signal and the response signal is used as the controller’s input, and the frequency is used as the output. The control system by the phase difference is maintained at around ±90° so that the test structure is always kept in the resonance state of a certain order mode [32]. 

The phase difference as the control input is obtained by the phase detector. In the phase detector of the control system, the two detected signals—input signal *f*_1_ of the shaking table and the test signal *f*_2_ of the resonant beam—have the same frequency.
(5)$$\begin{array}{l} f_{1}=A_{1}\sin\left(\omega t+\varphi_{1}\right)\\ f_{2}=A_{2}\sin\left(\omega t+\varphi_{2}\right) \end{array}$$

The signals *f*_1_ and *f*_2_ are input into the multiplier to obtain:(6)$$f_{1}\times f_{2}=\frac{1}{2}\times A_{1}\times A_{2}\times \left[\cos\left(\varphi_{1}-\varphi_{2}\right)-\cos\left(2\omega t+\varphi_{1}+\varphi_{2}\right)\right]$$

The output signal of the multiplier is low-pass filtered, leaving only the initial difference between the two signals. The output of the analog phase detector is related to the amplitude of the input signal. After removing the amplitude part, the phase difference between the two input signals can be obtained.

In Figure 7, we compare the amplitude–frequency characteristic curves and the phase–frequency characteristic curves measured under different accelerations. The resonant frequency of the resonant beam before increasing the acceleration is $f_{1}$. The phase difference $\varphi_{0}$ between the excitation signal and the output signal of the resonant beam sensor is obtained through the phase detector, and $\varphi_{0}$ at this time is phase-locked in the subsequent detection process.

The phase–frequency curve after increasing the acceleration is shown in Figure 5. The new phase difference is calculated as $\varphi_{2}$, and $\varphi_{2}$ is compared with $\varphi_{0}$. We control the excitation signal of the vibration device to change the frequency so that $\varphi_{2}$ and $\varphi_{0}$ are within the control interval. At this time, $f_{1}$ is the sensor’s new resonance frequency—the dashed line shown in Figure 5. Such repeated resonant beams will always work in a resonant state.

It is proved that there is a fixed relationship between the mechanical structure’s resonance peak and the input and output phase difference. It can be seen from Figure 5 that although the nonlinear change of amplitude and frequency is severe, it still maintains a consistent, continuous transition in the phase-frequency dimension. Thus, the control system can determine the frequency at which the resonance peak is located by observing the value of the phase difference [33].

Figure 8 visualizes a set of three-dimensional data on the resonance peak frequency–amplitude–phase difference measured at a length of 120 mm using a 5 mm Q235 steel resonant beam. Due to the considerable Q value of the mechanism, the peak interval of the frequency–amplitude curve is very narrow. This greatly increases the difficulty of frequency-based control. On the contrary, on the phase difference–amplitude curve, the curve near the peak is relatively flat. That is to say, by detecting the phase difference change, the control system can determine the deviation of the mechanical resonance peak value in a more timely manner. In addition, as long as the phase control is stable within a certain range, the mechanism is within the vicinity of the resonance peak, and the expected resonance excitation can be provided.

This article’s control system involves phase-locked resonance tracking, as shown in Figure 9. The system uses phase-detection-based dichotomy compensation [34]. After the system runs stably, each frequency tracking takes only 0.1 s, and there is no amplitude fluctuation, which meets the needs of vibration calibration.

## 3. Results

In order to verify the amplitude stability of the resonance control system, an experimental setup, as shown in Figure 10, was built. Where the standard sensor was a Endevco 2270 charge sensor, and the sensor used for testing is a Endevco 7240 charge sensor. Adopt Endevco’s signal amplifiers, NI’s sound and vibration module as control device. Standard sensor A was fixed at the installation position in the center of the resonant beam and was used to measure the acceleration of the resonant beam for use as a vibration control feedback signal. The calibration sensor and standard sensor A were connected “back-to-back.” At the same time, standard sensor B was installed on the vibration device table and was used to measure the acceleration of the vibration device table, serving as an auxiliary feedback signal for the vibration control system.

We measured the acceleration at the selected frequency, recorded the frequency and acceleration every 0.2 min, and measured it 12 times continuously. The acceleration stability of the system was calculated according to the following formula [30]:(7)$$S_{a}=\frac{2|\Delta a\left(i,i+1\right){|}_{\max}}{a_{i}+a_{i+1}}\times 100\%$$
where $S_{a}$ is the acceleration stability, %; ${\left|\Delta a\left(i,i+1\right)\right|}_{\max}$ is the maximum change in the acceleration value within 12 s, m/s^2^; and $a_{i},a_{i+1}$ are the two accelerations corresponding to the maximum change in the acceleration value within 12 s, m/s^2^.

As a measure of amplitude stability, S_*a*_ refers to ISO 16063-21 *Vibration calibration compared to a reference transducer*. We use this value to describe the acceleration stability of the system. The lower the value, the more stable the acceleration.

In order to make the experimental results reliable, the experiments were carried out under manual adjustment of the open-loop state, the original amplitude iterative algorithm control, and the control system described in this article.

Manual open-loop adjustment means that the amplitude and phase changes are constantly observed and adjusted before the start of the experiment so that the vibration amplitude of the mechanism is still in the resonant state when it reaches the target acceleration. Then, we stopped the adjustment to collect and record data. 

Figure 11 compares the amplitude stability with different control methods under amplitudes of 1000–8000 m/s^2^. It can be seen that when the acceleration is lower than 7000 m/s^2^, the amplitude stability of experiments is less than 2% for both methods. When the acceleration is greater than 7000 m/s^2^, the output amplitude of the vibration system using amplitude iterative control will oscillate continuously. The large and small amplitudes cause the acceleration stability to increase significantly. Furthermore, the stability under phase tracking control is less than 0.5%.

In Figure 12, we selected the more representative 4000 m/s^2^ data for comparison. The amplitude fluctuation of the phase tracking control—the blue line—is maintained within a small range. The amplitude control—the red line—still has timely and effective feedback control and its amplitude control is still stable, although the amplitude is offset. Feedback control was not performed for the control group (the black line). Although the amplitude fluctuation is insignificant, its acceleration amplitude deviates from the set value in the acquisition interval. It was used only as a reference for the amplitude fluctuation of the vibration device.

According to the analysis of the experimental data, it can be considered that the amplitude control method can still effectively control the acceleration at this stage. At this acceleration level, the resonant mechanism’s frequency shift is insignificant.

However, in Figure 11, the stability deviation of the amplitude control sharply increases when acceleration exceeds 7500 m/s^2^. We selected the experimental data at 8000 m/s^2^ for comparative analysis, as shown in Figure 13. It can be easily observed that in the acquisition interval, the amplitude control fluctuates wildly and increases continuously. At this point, the vibration control fails. The amplitude of each data point in the figure is the average value of the waveform collected for 12 s and calculated. Compared with 4000 m/s^2^, the amplitude shift of open-loop control is faster at 8000 m/s^2^. Offset for open loop control may be because the resonant mechanism works under high acceleration, and its internal temperature and assembly strength are affected, which changes the mechanism mode and affects the equivalent damping ratio ξ. This change gets faster as acceleration increases.

This is because, at such high accelerations, the frequency shift of the resonant mechanism is fast, and the bandwidth of the resonant peak is very small. When the resonance peak value of the mechanism is shifted, the magnification is dramatically reduced. The gain is reduced from g to g’, as shown in Figure 14. The feedback of the control system is excessive, and the amplitude control becomes an oscillatory process. Since these shifts happened all the time in the test, the system could no longer carry out stable control.

On the other hand, although the stability of open-loop control seems to be high, it can be seen from Figure 11 that its acceleration will gradually deviate from the set value, and the deviation is faster at high acceleration. 

Figure 15 shows the frequency–amplitude changes in the collection interval of the stability test experiment of the phase tracking method under 8000 m/s^2^. By tracking and locking the phase of the resonance state, the control system adjusts the frequency in time to follow the frequency offset. The amplitude variation of the resonant mechanism we can observe is relatively stable. This is because the calibration system can track the changing resonant frequency relatively quickly and smoothly, as shown by the smooth frequency adjustment curve in the figure. Therefore, the amplitude is affected very little.

The phase-locked frequency tracking control makes the gain of the resonance amplifier more stable. The acceleration change of the resonant beam is relatively stable; therefore, only slight amplitude control is required. It can be considered that the system can stably control the acceleration at the set amplitude.

## 4. Discussion

In this study, a novel phase-locked resonance tracking control method based on the phase resonance principle was proposed and applied to establish a set of resonant high-acceleration vibration automatic calibration systems. The high-acceleration sensor performance was calibrated via the phase resonance method to track the resonant frequency of the vibration system via the phase difference lock of the resonant mechanism. Standard sensors were installed on a vibration device resonant beam to monitor the resonance state when the system is near the resonant frequency. The vibration amplitude was stabilized by adjusting the output frequency to maintain the phase difference. Comparison experiments with the amplitude iteration method demonstrated that the amplitude stability with the proposed method was higher than that with the amplitude iteration method when the acceleration was greater than 7500 m/s^2^. The key method of the studied vibration control system is phase-locked tracking, which is stable and fast when compared to amplitude iteration. In the future, we plan to apply this system to calibrate the performance of high-acceleration vibration sensors.

## Figures and Tables

**Figure 1 sensors-22-07208-f001:**
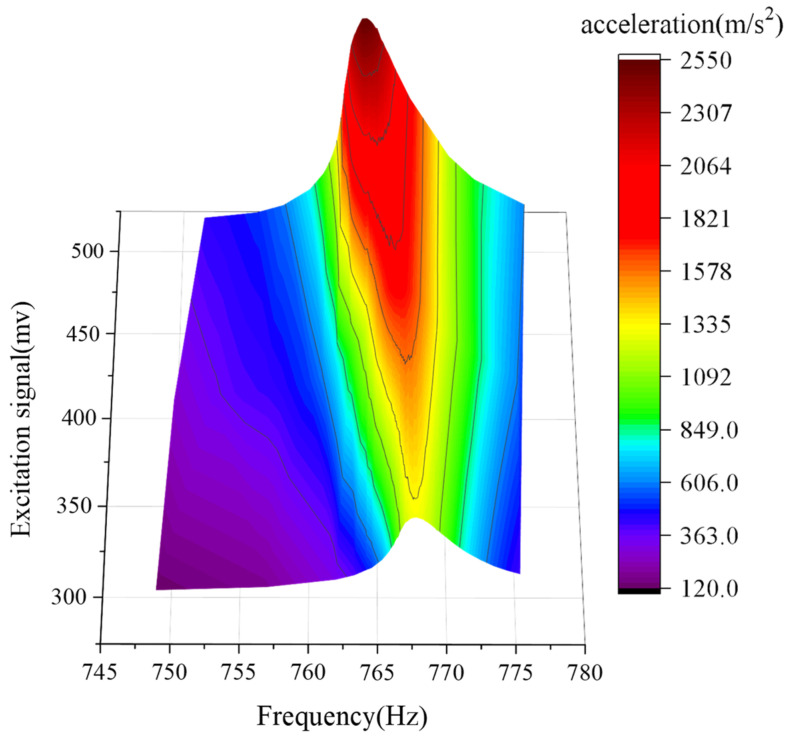
Schematic diagram of resonance peak shift.

**Figure 2 sensors-22-07208-f002:**
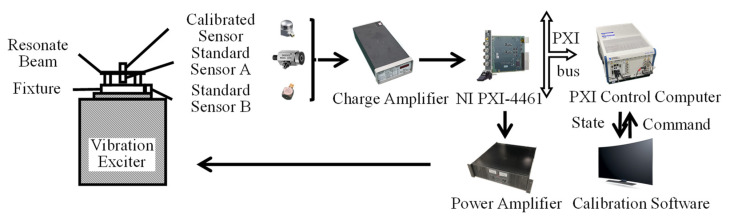
High-acceleration calibration device.

**Figure 3 sensors-22-07208-f003:**
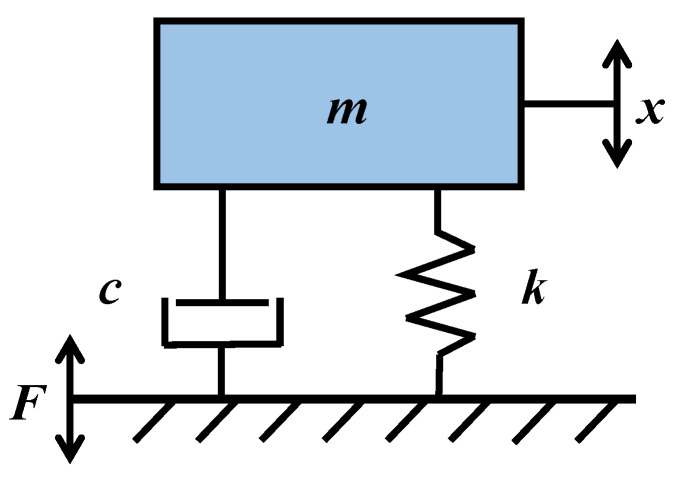
Simplified model of the resonant beam.

**Figure 4 sensors-22-07208-f004:**
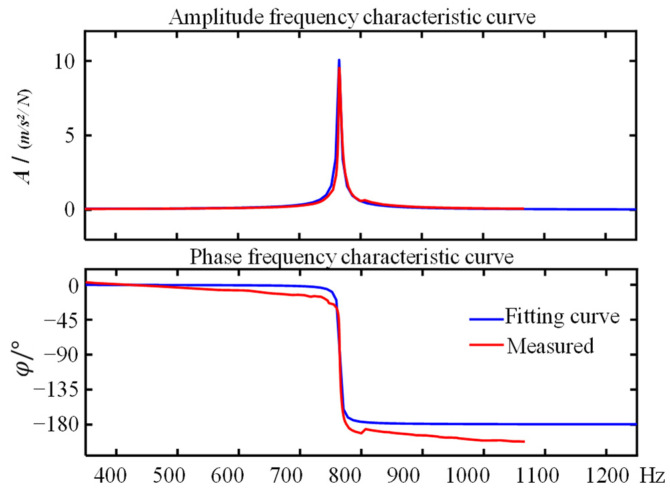
Fitting frequency characteristic curve of a resonance mechanism.

**Figure 5 sensors-22-07208-f005:**
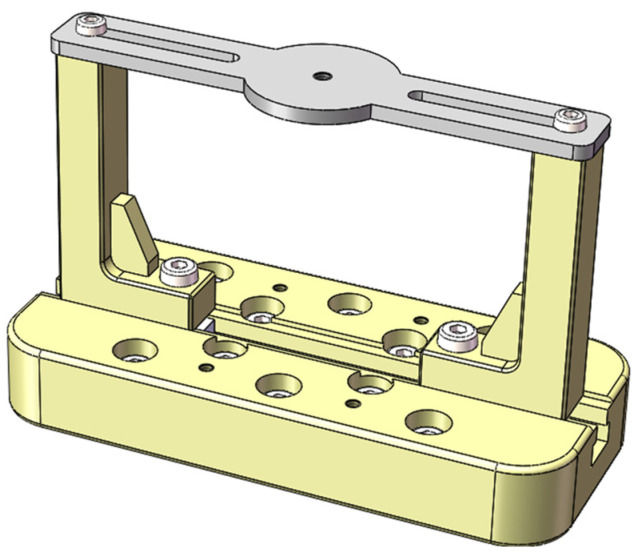
Simplified model of the resonant beam.

**Figure 6 sensors-22-07208-f006:**
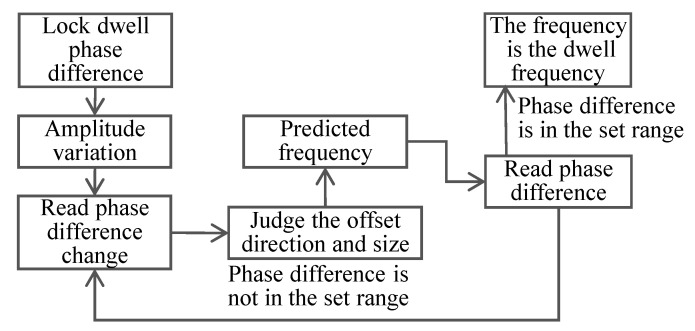
Iterative frequency resonance tracking method.

**Figure 7 sensors-22-07208-f007:**
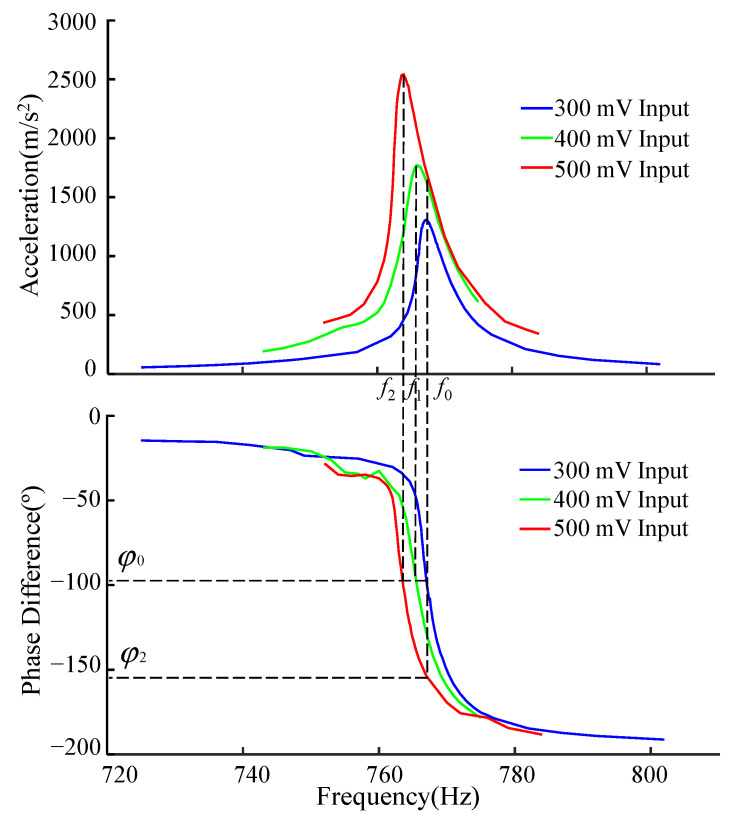
Amplitude–frequency and phase–frequency characteristic curves.

**Figure 8 sensors-22-07208-f008:**
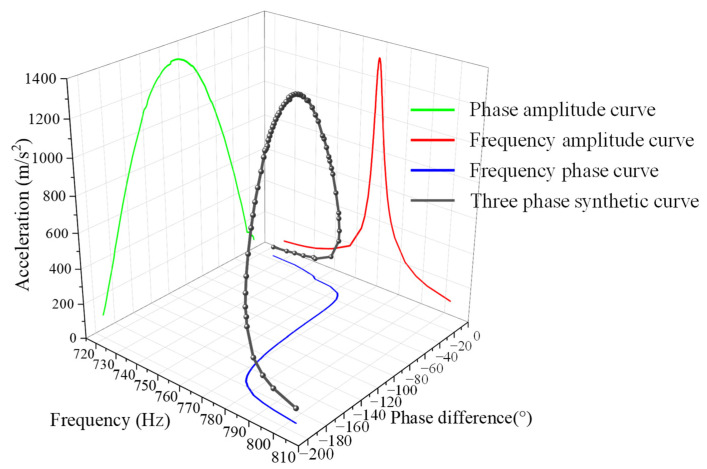
Phase–frequency change process.

**Figure 9 sensors-22-07208-f009:**
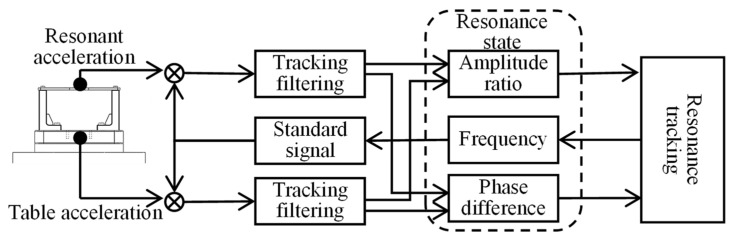
Phase-locked resonance tracking control.

**Figure 10 sensors-22-07208-f010:**
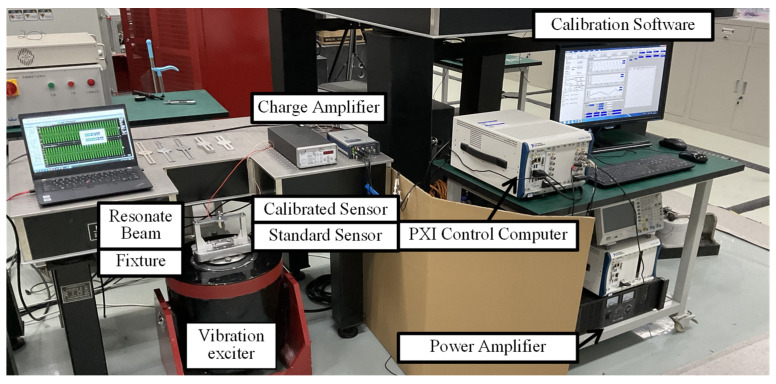
Resonant high-acceleration vibration control system.

**Figure 11 sensors-22-07208-f011:**
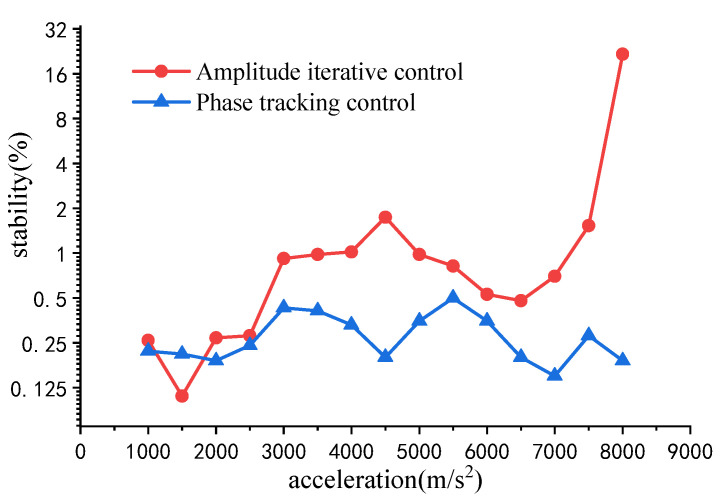
Amplitude stability comparison in the 1000–8000 m/s^2^ stage.

**Figure 12 sensors-22-07208-f012:**
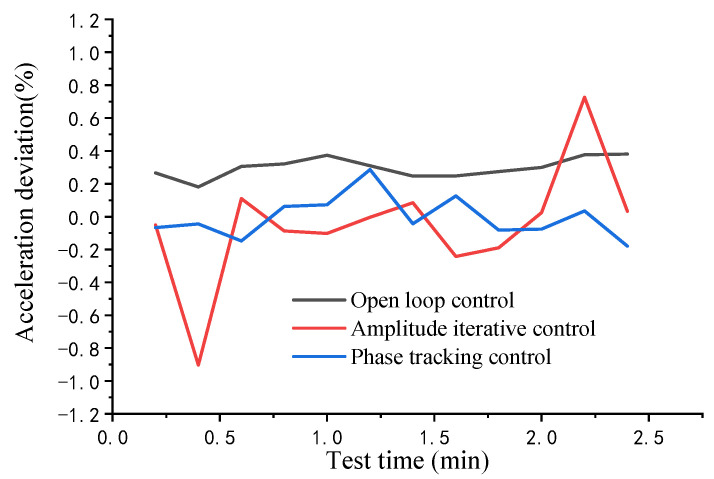
Amplitude stability comparison in the 4000 m/s^2^ interval.

**Figure 13 sensors-22-07208-f013:**
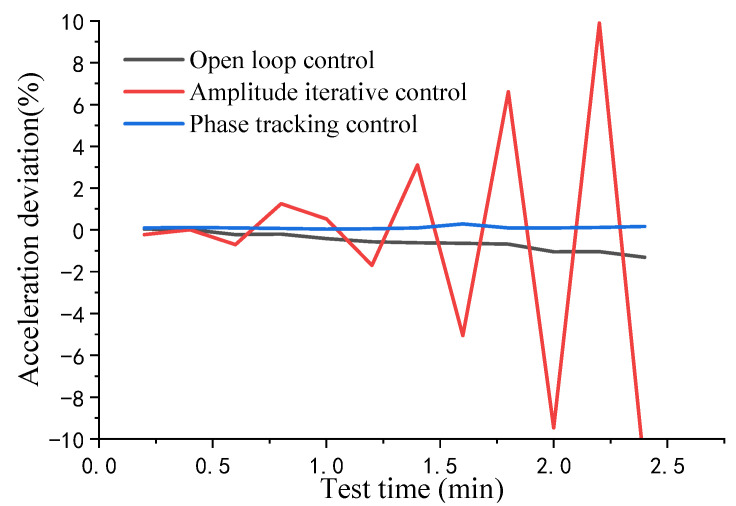
Amplitude stability comparison in the 8000 m/s^2^ interval.

**Figure 14 sensors-22-07208-f014:**
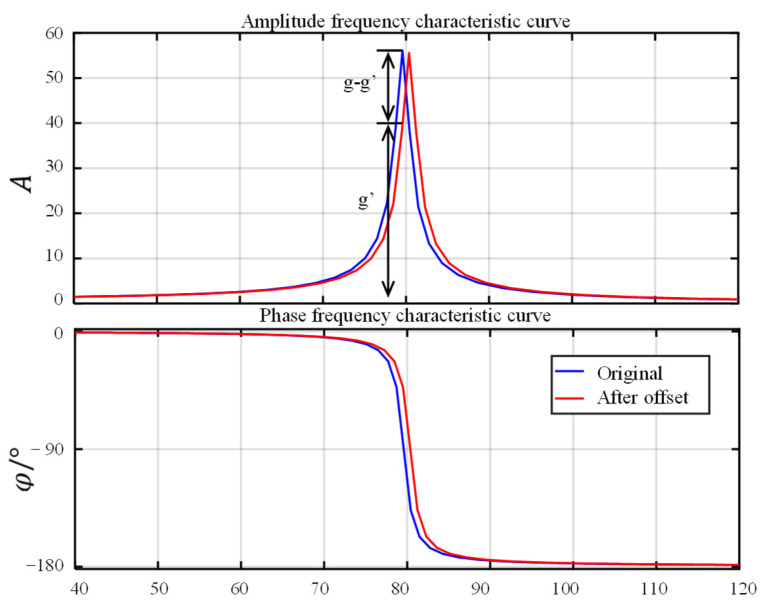
Schematic diagram of gain drop.

**Figure 15 sensors-22-07208-f015:**
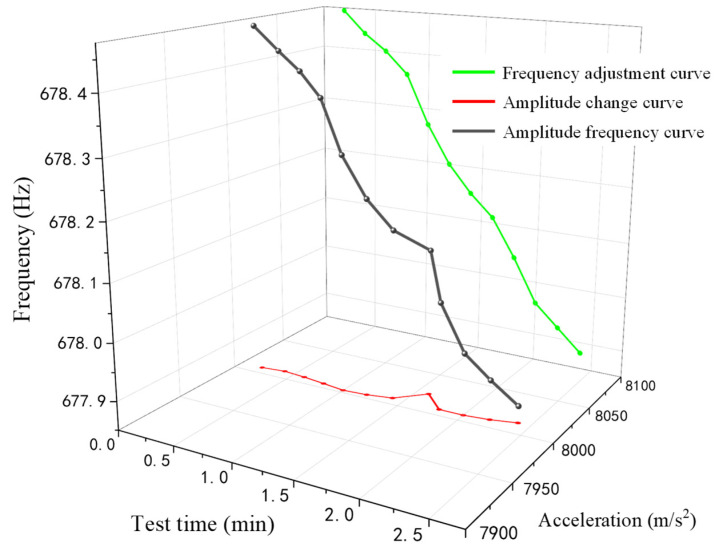
Schematic diagram of phase tracking amplitude control in the 8000 m/s^2^ interval.

**Table 1 sensors-22-07208-t001:** Main parameters of resonant beam material.

Material	Elastic Modulus E/GPa	Poisson’s Ratio *μ*	Density ρ/kg·m^−3^	Yield Strength *σ*_b_ /MPa
Q235	210	0.28	7800	220

## Data Availability

Not applicable.

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
