# Peer review of "High-Acceleration Vibration Calibration System Based on Phase-Locked Resonance Control"

_sensors, 2022, doi:10.3390/s22197208_

Round 1

Reviewer 1 Report

The high acceleration vibration sensor is employed in diverse applications ranging from structural science to aerospace engineering. This paper reports an approach that is based on the resonant phase lock technique, facilitating a relatively stable and precise control of the resonance amplification. Detailed experimental set-up and results are represented in this paper, along with a comparison with the conventional amplitude iterative control. This paper demonstrates a proof of concept work that the proposed resonant phase lock can be adopted in this specific context, and yield a solid functionality. Thanks for submitting this interesting work, it has the potential to be exploited in vibration sensors. However, there are some comments that have to be addressed.

Comments:

1.      Concerning the whole text, the English writing is a little brusque, some statements are inconsistent, or disjointed. I suggest a careful revision regarding the interpretation and representation, in particular, the introduction section.

2.      The Keywords, in line 25, need to be adjusted. The vital part of this work is the resonant phase lock, for a specific application of the resonant beam, that is utilized in vibration acceleration. Hence, the vibration sensor itself is not the major content, actually, the author used standard vibration sensors (commercially available?).

3.      As an example of the English writing problem, in line 34, the author states that a vibration table is an effective strategy to calibrate sensors, but why in the subsequent sentence “it is imperative to develop an applicable vibration calibration method” Does the author want to say the vibration table incorporates with dedicated vibration calibration technique to be served as a possible calibration method for the high-acceleration sensor?

4.      Line 57, regarding the large acceleration above 10000 m/s^2, the author wants to state the crucial problem of such a large gain output from the resonant beam, but the content is difficult for the reader to understand, thus a rephrase is demanded.

5.      Equation (1), line 91, seems not correct. The equation of motion is spring-mass-damper system under an excitation force. However, m*X is not a force, there should be m*acceleration, author please check. The negative sign here is also strange, or an interpretation is needed.

6.      Equation (2), line 95, seems not correct. The numerator should be m, hence the subsequent expression is valid.

7.      Equation (3), line 97, the author should clarify how the term wn2 in numerator comes. The transfer function is obtained in Equ. (2), here it should be the amplification gain of resonance G(s). An explanation is required.

8.      In figure 3, there is no unit on Y-axis, the amplitude A should be given a proper unit.

9.      Equation (4) is kind of vague, this is a calculated transfer function from the fitting. However, basic parameters of the resonant beam, such as effective mass and stiffness, should be given, and use equation (3) to calculate the frequency-amplitude and phase-frequency responses.

10.   Referring to the resonant beam, the author should provide a schematic of the beam structure (3D graph), along with some physical parameters of the beam, for example, the Q-factor in the air (I suppose this is the experimental environment), the material of the beam, the dimensions etc, can be summarized in a table.

11.   In line 151, f1 is mentioned, however, it is absent in figure 5.

12.   Based on figure 5, non-linear behaviour is observed when the excitation signal is in the hundreds of mV range. This non-linear response causes an unsymmetrical frequency-amplitude response, and this is already obvious with the output acceleration of less than 3000 m/s2. If it is possible, the author should give a frequency response, say with an acceleration larger than 7500 m/s2, as this large acceleration output is a feature of this work and it could cause a heavy nonlinearity. Heavy nonlinearity will affect the frequency-phase and frequency-amplitude drastically, thus there might be a critical problem with the phase lock. The author should clarify such a context.

13.   In Figure 2 and Figure 8, specific facilities such as the charge amplifier, sensors and power amplifier are commercialized production, please give more information about them. (brand, type etc)

14.   Equation (7) and Figure 9, the Sa should be instability, as well as the label of the Y-axis in Figure 9. Also the statements in the relevant paragraph, the stability should change to instability.

15.   The author should clarify the frequency offset due to open loop control, giving a proper explanation that dissects the possible reasons. For instance, the frequency drifting of the resonant beam, the ambient temperature fluctuations or pressure changes, and the error caused by full frequency sweep and record facilities.

16.   The expression “peak range” is vague, the author wants to explain the bandwidth of the resonant peak, please rephrase the words.

17.   In Figure 13, the frequency adjustment curve follows exactly the same trends as the A-F curve, it seems wrong. I think the adjustment should be carried out to compensate for the frequency change due to input, thus the curve should be the opposite trend compared to the A-F, author please check and clarify.

18.   All the experiment time seems short (a few minutes length), what happens if the experiment time reaches tens of minutes, an hour or a few hours?

19.   If the author would like to further refine the experimental results, a noise analysis (open loop, amplitude iteration, phase lock) along with the Allan-deviation method to determine the stability, can improve the scientific contents of the submitted manuscript.

20.   If the phase lock is applied, the system is considered a closed-loop, and the tracking time, as mentioned by the author, is less than 0.1s. How good is this performance compared to other state-of-the-art, and in practical cases, how to tackle the problem if the phase variation is faster than 0.1s?

21.   Reference 1 is not correct.

Author Response

Dear reviewer:

I am very grateful to your comments for our manuscript, we amended the relevant part in manuscript. Some of your questions were answered below. (This response is also attached at the end of the attachment.)

Comment #1: Concerning the whole text, the English writing is a little brusque, some statements are inconsistent, or disjointed. I suggest a careful revision regarding the interpretation and representation, in particular, the introduction section.

The authors’ Answer: MDPI's native English-speaking editors have been commissioned to polish the English writing and revise the Introduction. A language polish certificate is attached at the end.

Comment #2: The Keywords, in line 25, need to be adjusted. The vital part of this work is the resonant phase lock, for a specific application of the resonant beam, that is utilized in vibration acceleration. Hence, the vibration sensor itself is not the major content, actually, the author used standard vibration sensors (commercially available?).

The authors’ Answer: First, the keyword was adjusted according to the reviewer's suggestion; second, whether a standard or commercial sensor is available, this standard sensor can be used to calibrate the commercial sensor.

Comment #3: As an example of the English writing problem, in line 34, the author states that a vibration table is an effective strategy to calibrate sensors, but why in the subsequent sentence “it is imperative to develop an applicable vibration calibration method” Does the author want to say the vibration table incorporates with dedicated vibration calibration technique to be served as a possible calibration method for the high-acceleration sensor?

The authors’ Answer: It more precisely means "an applicable vibration control method must be developed" The shaker is used as part of a calibration system to which this method is applied, to provide high accelerations to calibrate the corresponding sensors. The formulation in this section has been corrected in the original text.

Comment #4: Line 57, regarding the large acceleration above 10,000 m/s^2, the author wants to state the crucial problem of such a large gain output from the resonant beam, but the content is difficult for the reader to understand, thus a rephrase is demanded.

The authors’ Answer: The meaning here is that the vibration above 7,000 m/s^2 cannot be controlled stably, affecting the calibration ability. At present, commercial sensors have reached an acquisition capacity of 10,000 m/s^2. There is an urgent need to improve calibration capacity to meet commercial and academic needs. The formulation in this section has been corrected in the original text. Please pay attention to whether the statement is clear.

Comment #5: Equation (1), line 91, seems not correct. The equation of motion is spring-mass-damper system under an excitation force. However, m*X is not a force, there should be m*acceleration, author please check. The negative sign here is also strange, or an interpretation is needed.

The authors’ Answer: We are very sorry for our negligence. The X should be replaced by X¨, and the equation will be correct. However, considering that the input signal of the excitation force of the vibration device was used in our data collection, it is more appropriate to express this equation as force F. Equation 1 has been corrected in the original text.

Comment #6: Equation (2), line 95, seems not correct. The numerator should be m, hence the subsequent expression is valid.

The authors’ Answer: We are very sorry for our negligence. The new equation 2 has been re-derived from the new equation 1.

Comment #7: Equation (3), line 97, the author should clarify how the term wn2 in numerator comes. The transfer function is obtained in Eq. (2), here it should be the amplification gain of resonance G(s). An explanation is required.

The authors’ Answer: This was a bug, so we fixed the formula. G(s) is the change from the excitation force of the vibration device to the acceleration gain, and the derivation process is added in the original text for explanation.

Comment #8: In figure 3, there is no unit on Y-axis, the amplitude A should be given a proper unit.

The authors’ Answer: We are very sorry for our negligence. The amplitude A here refers to the gain multiple of the structure. The unit of the amplitude A should be m/s2/N.

Comment #9: Equation (4) is kind of vague, this is a calculated transfer function from the fitting. However, basic parameters of the resonant beam, such as effective mass and stiffness, should be given, and use equation (3) to calculate the frequency-amplitude and phase-frequency responses.

The authors’ Answer: The equivalent damping ratio coefficient ξ in the original formula (3) will be affected by the resonance mechanism's assembly and acceleration, making it difficult to calculate. On the other hand, since the resonance mechanism is a part of the final output acceleration of the system, the input voltage signal also needs to go through the power amplifier for gain. Therefore, the paper does not take a mathematical method to calculate the transfer function of a specific assembly. A more straightforward fitted transfer function is presented here to verify that the experimental structure conforms to the expected design, which can be applied to the control system under study.

At the same time, your suggestion is significant. The basic schematic and parameters of the resonance mechanism have been supplemented in the manuscript, making the overall physical model more specific.

Comment #10: Referring to the resonant beam, the author should provide a schematic of the beam structure (3D graph), along with some physical parameters of the beam, for example, the Q-factor in the air (I suppose this is the experimental environment), the material of the beam, the dimensions etc, can be summarized in a table.

The authors’ Answer: We have added the 3D map of the resonant beam and provided its associated data. The quality Q-factor here refers to the sharpness of the resonance peak, which is generally used to describe the amplification ability of the gain system.

Comment #11: In line 151, f1 is mentioned, however, it is absent in figure 5.

The authors’ Answer: Thanks for your suggestion, we have added f1 on figure 5

Comment #12: Based on figure 5, non-linear behavior is observed when the excitation signal is in the hundreds of mV range. This non-linear response causes an unsymmetrical frequency-amplitude response, and this is already obvious with the output acceleration of less than 3000 m/s2. If it is possible, the author should give a frequency response, say with an acceleration larger than 7500 m/s2, as this large acceleration output is a feature of this work and it could cause a heavy nonlinearity. Heavy nonlinearity will affect the frequency-phase and frequency-amplitude drastically, thus there might be a critical problem with the phase lock. The author should clarify such a context.

The authors’ Answer: At accelerations as high as 7500 m/s², the resonant frequency shifts very rapidly. It is difficult for us to quickly record the magnitude response of different frequencies. Therefore, we cannot draw the amplitude-frequency characteristic curve at a certain moment.

On the other hand, it can be seen from Figure 5 that although the nonlinear change of amplitude and frequency is severe, it still maintains a consistent, continuous transition in the phase-frequency dimension. Based on the consistency of the phase change and the pointing effect on the formant, this paper proposes to use phase-locked control to improve the control capability under high acceleration. We will add to this in the text.

Comment #13: In Figure 2 and Figure 8, specific facilities such as the charge amplifier, sensors and power amplifier are commercialized production, please give more information about them. (brand, type etc)

The authors’ Answer: We have provided information on the model and main parameters of the sensors and signal amplifiers used.

Comment #14: Equation (7) and Figure 9, the Sa should be instability, as well as the label of the Y-axis in Figure 9. Also the statements in the relevant paragraph, the stability should change to instability.

The authors’ Answer: Generally, in practice we call Sa as stability. Refer to ISO 16063-21 Vibration calibration compared to a reference transducer. Sa as an indicator, we use it to describe the acceleration stability of the system. The lower the value, the higher the stability. We found that such a description would indeed lead to many misunderstandings, so we made a supplementary explanation in the text.

Comment #15: The author should clarify the frequency offset due to open loop control, giving a proper explanation that dissects the possible reasons. For instance, the frequency drifting of the resonant beam, the ambient temperature fluctuations or pressure changes, and the error caused by full frequency sweep and record facilities.

The authors’ Answer: In the experimental results section, we used the results of the open-loop control as a control group to support the necessity of control. As we mentioned in the background, the resonant frequency of the mechanism shifts as the acceleration changes. The input signal is fixed at the initial resonant frequency in open-loop control. With the shift of the resonant frequency, the gain multiple changes, and the output acceleration of the open-loop control gradually deviates from the preset value. We believe this offset may be because the resonant mechanism works under high acceleration, and its internal temperature and assembly strength are affected, which changes the mechanism mode and affects the equivalent damping ratio ξ. This part is less elaborated on in the original article, and we have added it to the Discussion section.

Comment #16: The expression “peak range” is vague, the author wants to explain the bandwidth of the resonant peak, please rephrase the words.

The authors’ Answer: We believe that the bandwidth of the formants can be expressed more clearly, they have been corrected in this revision manuscript.

Comment #17: In Figure 13, the frequency adjustment curve follows exactly the same trends as the A-F curve, it seems wrong. I think the adjustment should be carried out to compensate for the frequency change due to input, thus the curve should be the opposite trend compared to the A-F, author please check and clarify.

The authors’ Answer: The output frequency of the resonant mechanism we can observe is the same as the excitation frequency. So, the frequency curve here is consistent with the A-F curve projection on the frequency dimension. We can only show the effect of tracking the expected resonant frequency by how stable the amplitude is. This "frequency adjustment curve" actually means that the calibration system can track the changing resonant frequency relatively quickly and smoothly, with less impact on the amplitude. We have revised the instructions here to make the article easier to understand.

Comment #18: All the experiment time seems short (a few minutes length), what happens if the experiment time reaches tens of minutes, an hour or a few hours?

The authors’ Answer:

 There are a few things I would like to explain to you about the time length of the experiment:

  1. The experimental results shown in the article are selected from a large number of experimental results in order to facilitate comparison and make it easy for readers to understand after being displayed. Of course, their trends are consistent with the experimental conclusions. For example, in Figure 10, you can observe that the open loop control deviation curve does not start at 0% but is slightly higher. Because the amplitude shift is insignificant under this acceleration, we selected the data after the system ran for 20 minutes for display.
  2. Figure 11 shows that amplitude control is already precarious under this acceleration. If the experiment is carried out for a long time, it will inevitably cause adverse effects on the equipment. Furthermore, if the curve with too large a value is displayed, the result of the open-loop test will be small and difficult to observe.
  3. On the other hand, our standard sensors all have suitable operating frequency ranges. Longer operating times will cause the vibration frequency deviation to be too large, and the basic meaning of calibration will be lost. Of course, We can assure you that our system will run stably within the time required for calibration.

Comment #19: If the author would like to further refine the experimental results, a noise analysis (open loop, amplitude iteration, phase lock) along with the Allan-deviation method to determine the stability, can improve the scientific contents of the submitted manuscript.

The authors’ Answer: Thanks for your valuable advice. We know that the experimental results of this article may be relatively straightforward. Therefore, we are also conducting more detailed comparative experiments based on this system. Including uncertainty, lateral vibration ratio, etc., to test vibration stability in the different methods you mentioned. We hope future work can use these experimental data better to analyze the scientific content in mechanism resonance control. Now, this article focuses more on introducing the construction and case of a stable high acceleration calibration system to solve the more urgent measurement needs.

Comment #20: If the phase lock is applied, the system is considered a closed-loop, and the tracking time, as mentioned by the author, is less than 0.1s. How good is this performance compared to other state-of-the-art, and in practical cases, how to tackle the problem if the phase variation is faster than 0.1s?

The authors’ Answer: As a closed loop system, it considers the vibration device's service life. The control system is designed with a delay adjustment of 0.075s per cycle to take care of the computer's response time and the ability to vibrate the device to avoid oscillation. For this tracking time, the frequency offset caused by the 14,000 m/s^2 tried in the experimental setup is stably controllable. It can also be observed from the original Figure 13 that the system does not cause a significant change in amplitude due to the long tracking time. We have tried a tracking time of 0.2s and can still achieve stable control at 10,000 m/s^2. As an innovative work, we have no way of knowing whether this tracking time is better than similar advanced products, but it fully meets the needs of existing experiments.

Comment #21: Reference 1 is not correct.

The authors’ Answer: We are very sorry for our negligence. This was a formatting error, sorry it has been corrected.

Thank you for your advice. It is very important. Due to your suggestions, I have discovered the deficiencies in my current work. I will follow your suggestions in future work to improve the level of scientific research and achieve more!

Thank you again for your valuable suggestions!

Sincerely yours,

Ran Cheng

Reviewer 2 Report

This work examined the phase-locked resonance tracking control method based on the proposed phase resonance principle, which can improve the accuracy and stability of resonance control. Furthermore, it can apply to establish resonant high acceleration vibration automatic calibration systems. The high-acceleration sensor performance is calibrated using the phase resonance method. This work is impressive; I suggest this research article can be published in the journal after a minor revision.

1.   The parameters in equations 3 are not clearly defined; for example, G(s).

2.  The acceleration stability of the formula (Eqn. 7) must cite the reference.

3.     Figure 9 shows that when the acceleration is less than 7000 m/s2, the amplitude stability of experiments is less than 2%. When the acceleration is more than 7000 m/s2, the acceleration stability significantly increases up to ~32. The author has to describe the phenomena.

4.       Kindly enhance the ratio of the latest references, for examples:

 i.        A calibration system of resonant high-acceleration and metrological traceability [Meas. Sci. Technol. 32, 125904, (2021)]

ii.  Primary accelerometer calibration by scanning laser Doppler vibrometry [Meas. Sci. Technol. 31, 065006, (2020)]

iii.   Monocular vision-based Earth's graviation method used for low-frequency vibration calibration [IEEE Access 8, 129087–129093, 2020].

5. Metamaterials are artificial materials that have strong electromagnetic resonance. The resonance frequencies of metamaterials depend on their geometrical structures. [Photonics Research 9(10), 1970-1978, (2021); Photonics Research 9(7) 1409-1415, (2021); Photonics Research 9(2), 125-130 (2021)] Is it possible to use metamaterials to develop advanced vibration sensors? Please comment these references in manuscript.

Author Response

Dear reviewer:

I am very grateful to your comments for our manuscript, we amended the relevant part in manuscript. Some of your questions were answered below. (This response is also attached at the end of the attachment.)

Comment #1: The parameters in equations 3 are not clearly defined; for example, G(s).

The authors’ Answer: G(s) is the transfer function between the input excitation force and the mechanism output acceleration. There are some errors in the original formula. The equations have been re-derived and precisely defined.

Comment #2: The acceleration stability of the formula (Eqn. 7) must cite the reference.

The authors’ Answer: The acceleration stability of the formula (Eqn. 7) referred to ISO 16063-21 Vibration calibration by comparison to a reference transducer. We have added the explanation and citation in this revision manuscript.

Comment #3: Figure 9 shows that when the acceleration is less than 7000 m/s2, the amplitude stability of experiments is less than 2%. When the acceleration is more than 7000 m/s2, the acceleration stability significantly increases up to ~32. The author has to describe the phenomena.

The authors’ Answer: When the acceleration is more excellent than 7000m/s2, the output amplitude of the vibration system using amplitude iterative control will oscillate continuously. The large and small amplitudes cause the acceleration stability to increase significantly. Phenomenon descriptions in this section are added to the text. The reason for this is briefly discussed in Figure 12.

Comment #4: Kindly enhance the ratio of the latest references.

The authors’ Answer: Appropriately added up-to-date references based on your suggestions.

Comment #5: Metamaterials are artificial materials that have strong electromagnetic resonance. The resonance frequencies of metamaterials depend on their geometrical structures. [Photonics Research 9(10), 1970-1978, (2021); Photonics Research 9(7) 1409-1415, (2021); Photonics Research 9(2), 125-130 (2021)] Is it possible to use metamaterials to develop advanced vibration sensors? Please comment these references in manuscript.

The authors’ Answer: Thank you very much for your suggestion. We read these papers carefully and found that the related research on Metamaterials is quite enlightening for our research work. The fundamental cause of the resonance shift is the change in the resonance frequency of the mechanical structure. One way to suppress the frequency shift is to reduce the influence of factors other than the geometric structure on the resonant frequency. Different designs of metamaterials exhibit resonant properties, which have a positive effect on the development and calibration of high-acceleration vibration sensors. We have added relevant comments to the manuscript and hope to try related applications of Metamaterials in future research.

Thank you for your advice. It is very important. Due to your suggestions, I have discovered the deficiencies in my current work. I will follow your suggestions in future work to improve the level of scientific research and achieve more!

Thank you again for your valuable suggestions!

Sincerely yours,

Ran Cheng

Round 2

Reviewer 1 Report

Thank you for the great effort of revising the manuscript.

The contents of the revised version are improved signifiantly.

There is just a minor error, line 223, an additional "bandwidth" appears. 

Author Response

Dear reviewer:
According with your advice, we amended the relevant part in manuscript. Thank you again for your valuable suggestions! 
Your review is very important to our research. I will follow your suggestions in future work to improve the level of scientific research and achieve more!

Thank you for acknowledging our work.
Sincerely yours,
RanCheng